# A Comprehensive Review on Bacterial Vaccines Combating Antimicrobial Resistance in Poultry

**DOI:** 10.3390/vaccines11030616

**Published:** 2023-03-08

**Authors:** Md. Saiful Islam, Md. Tanvir Rahman

**Affiliations:** Department of Microbiology and Hygiene, Faculty of Veterinary Science, Bangladesh Agricultural University, Mymensingh 2202, Bangladesh

**Keywords:** bacterial vaccines, antimicrobial resistance, poultry, bacterial diseases, salmonellosis, avian colibacillosis, disease prevention, global poultry sector

## Abstract

Bacterial vaccines have become a crucial tool in combating antimicrobial resistance (AMR) in poultry. The overuse and misuse of antibiotics in poultry farming have led to the development of AMR, which is a growing public health concern. Bacterial vaccines are alternative methods for controlling bacterial diseases in poultry, reducing the need for antibiotics and improving animal welfare. These vaccines come in different forms including live attenuated, killed, and recombinant vaccines, and they work by stimulating the immune system to produce a specific response to the target bacteria. There are many advantages to using bacterial vaccines in poultry, including reduced use of antibiotics, improved animal welfare, and increased profitability. However, there are also limitations such as vaccine efficacy and availability. The use of bacterial vaccines in poultry is regulated by various governmental bodies and there are economic considerations to be taken into account, including costs and return on investment. The future prospects for bacterial vaccines in poultry are promising, with advancements in genetic engineering and vaccine formulation, and they have the potential to improve the sustainability of the poultry industry. In conclusion, bacterial vaccines are essential in combating AMR in poultry and represent a crucial step towards a more sustainable and responsible approach to poultry farming.

## 1. Introduction

Poultry production is a critical component of the global food industry, providing a low-cost source of protein to millions of people worldwide [1]. However, the poultry industry is facing a significant challenge in the form of antimicrobial resistance (AMR) [2]. AMR refers to the ability of bacteria to resist the effects of antibiotics, and it is driven, in part, by the overuse and misuse of antibiotics in poultry production [3]. The widespread use of antibiotics in poultry production has led to the emergence of antibiotic-resistant bacteria, which can be transmitted to humans through the food chain and cause serious health problems [4]. AMR is a growing public health concern, as it reduces the effectiveness of antibiotics, making it harder to treat bacterial infections [5].

Bacterial vaccines offer a promising strategy for combating AMR in poultry production. Unlike antibiotics, which are broad-spectrum drugs that can target both pathogenic and non-pathogenic bacteria, bacterial vaccines are specific to a particular pathogen and do not have the same impact on the gut microbiome [6]. This means that they are less likely to contribute to the development of AMR, making them a more sustainable alternative to antibiotics. Bacterial vaccines work by stimulating the poultry’s immune system to produce a response against specific bacterial pathogens, reducing the need for antibiotics. They offer long-lasting protection against bacterial infections, reducing the need for repeated treatments with antibiotics [7]. Despite the benefits of using bacterial vaccines in poultry, there are also limitations to their use. The use of bacterial vaccines in poultry is regulated by various governmental agencies, and there are economic considerations to be taken into account, including costs and return on investment [8]. Additionally, there is an ongoing debate about the efficacy of bacterial vaccines, and some studies have shown that vaccine efficacy can vary depending on the target bacteria and the specific vaccine used [9,10].

Bacterial vaccines play a critical role in combating the growing problem of AMR in poultry and represent a step towards a more sustainable and responsible approach to poultry farming. With continued research and innovation, bacterial vaccines have the potential to greatly improve the health and welfare of poultry, as well as protect public health by reducing the spread of AMR. The purpose of this review article is to provide a comprehensive overview of the use of bacterial vaccines fighting AMR in poultry, including the benefits and limitations of their use, and to discuss future prospects for their development and use. The article also provides a critical evaluation of the literature on bacterial vaccines in poultry, highlights the key challenges and opportunities in this field, and delivers recommendations to the poultry authorities about the importance of bacterial vaccines instead of antimicrobial use to combat AMR and its consequences in poultry production.

## 2. Overview of Bacterial Diseases in Poultry

Bacterial diseases in poultry can cause significant morbidity and mortality, leading to economic losses for farmers. Some of the most important bacterial diseases in poultry include the following [11]:Salmonellosis: A bacterial infection caused by *Salmonella*, which can lead to severe diarrhea, septicemia, and death in poultry.Colibacillosis: A bacterial infection caused by *Escherichia coli*, which can cause severe diarrhea in young chicks and septicemia in adult chickens.Avian Mycoplasmosis: A bacterial infection caused by *Mycoplasma gallisepticum*, which affects the respiratory and reproductive systems of poultry and can cause decreased egg production and increased mortality.Pasteurellosis: A bacterial infection caused by *Pasteurella multocida*, which affects the respiratory system of poultry and can cause severe pneumonia, septicemia, and death.Campylobacteriosis: A bacterial infection caused by *Campylobacter jejuni*, which can cause severe diarrhea, enteritis, and septicemia in poultry.*Staphylococcus* infection: A bacterial infection caused by *Staphylococcus aureus*, which can cause skin and wound infections, arthritis, and septicemia in poultry.Chlamydiosis: A bacterial infection caused by *Chlamydia psittaci*, which affects the respiratory and reproductive systems of poultry and can cause decreased egg production and increased mortality.

Common bacterial diseases developed in poultry are detailed in Table 1.

Bacterial diseases in poultry have significant impacts on the poultry industry. The diseases can cause high mortality rates in poultry, decreased egg production, and reduced quality of meat and eggs. These losses can result in significant financial losses for poultry farmers and the poultry industry as a whole [15]. Additionally, bacterial diseases in poultry can lead to increased use of antibiotics, contributing to the development of antibiotic resistance.

Controlling bacterial diseases in poultry is crucial for the health and productivity of poultry and for the profitability of poultry production. Effective control measures can include biosecurity measures, such as good sanitation practices, controlling the movement of poultry, and preventing the introduction of infected chickens into a flock. Additionally, vaccination programs can play an important role in controlling bacterial diseases in poultry, providing an effective and cost-efficient way to prevent or reduce the impact of bacterial diseases in poultry.

## 3. Antibiotics Contribute to Antimicrobial Resistance

Antibiotics are powerful medicines that have revolutionized modern medicine by providing a way to treat bacterial infections that were once life-threatening. However, their widespread use has led to the development of antibiotic resistance, which is one of the most significant threats to global public health today [16,17]. The following are some ways in which antibiotics contribute to the development of antimicrobial resistance [18,19,20,21,22]:Overuse and misuse of antibiotics: Overuse and misuse of antibiotics are the primary causes of antibiotic resistance. Antibiotics are often prescribed for viral infections that they cannot cure, leading to unnecessary exposure to antibiotics, and making it easier for bacteria to develop resistance. Moreover, people often stop taking antibiotics once they feel better, not realizing that the bacteria may still be present, leading to incomplete treatment and the development of resistance.Selection pressure: Antibiotics exert strong selection pressure on bacteria, killing off the susceptible ones and allowing the resistant ones to survive and multiply. The resistant bacteria then go on to spread their resistance genes to other bacteria through horizontal gene transfer, including plasmids, transposons, and integrons. This horizontal transfer of resistance genes can occur within and between different species, making it harder to control the spread of resistance.Antibiotic residues in the environment: Antibiotics and their metabolites can persist in the environment for a long time after they are used, even at low concentrations. This can lead to the selection of resistant bacteria in the environment, which can then spread to humans and animals. Antibiotic residues in water bodies can also contribute to the spread of resistance, as they can lead to the selection of resistant bacteria in aquatic environments.Agricultural use of antibiotics: Antibiotics are widely used in agriculture, both to treat and prevent infections in animals and as growth promoters. This can lead to the selection of resistant bacteria in animals, which can then spread to humans through the food chain or the environment. Moreover, the use of antibiotics in agriculture contributes to the spread of resistance by releasing antibiotic residues into the environment.Lack of new antibiotics: The development of new antibiotics has slowed down in recent years, partly due to the high cost and time required for the development. This means that the antibiotics we have now are becoming less effective against resistant bacteria, which can lead to the further spread of resistance.

## 4. Antimicrobial Resistance in Poultry

AMR refers to the ability of microorganisms, including bacteria, viruses, fungi, and parasites, to resist the effects of antimicrobial drugs that were previously effective in treating infections caused by these microorganisms. AMR is a growing global public health threat and a major concern in the poultry industry [23], as poultry are often treated with antibiotics to prevent and control bacterial diseases. Moreover, antimicrobial-resistant bacteria in poultry can pose a risk to human health through the consumption of contaminated poultry products [24].

There are several mechanisms by which bacteria can develop resistance to antibiotics [25], including the following:Mutations: Bacteria can undergo genetic mutations that alter their structure and make them resistant to antibiotics.Horizontal gene transfer: This occurs when bacteria transfer genes that confer resistance to antibiotics to other bacteria through mechanisms such as conjugation, transduction, and transformation.Antibiotic pressure: The overuse and misuse of antibiotics can select bacteria that are resistant to these drugs.

The development of AMR in poultry has significant impacts on the poultry and poultry industry [26], including the following (Figure 1):Spread of resistance: Antibiotic-resistant bacteria can spread from poultry to poultry through direct contact or the environment. This can contribute to the spread of resistance and make it more challenging to control bacterial infections in poultry. Moreover, resistance genes can be transferred from bacteria in poultry to bacteria in other poultry through horizontal gene transfer, leading to the further spread of resistance.Reduced efficacy of antibiotics: Antibiotic-resistant bacteria are more difficult to treat and can lead to prolonged illness and increased mortality in poultry.Impact on animal welfare: The use of antibiotics in poultry can contribute to the development of antibiotic-resistant bacteria, which can lead to the increased use of antibiotics and the use of higher doses, leading to the potential for adverse effects on animal welfare.Increased costs: The treatment of antibiotic-resistant infections can be more expensive and time-consuming than the treatment of infections caused by susceptible bacteria.Reduced productivity: Antimicrobial-resistant infections can reduce the health and productivity of poultry flocks, leading to decreased egg production and reduced meat quality.

## 5. Overview of the History of Bacterial Vaccines in Poultry

Bacterial vaccines have been used in poultry production for several decades [27]. Since then, the use of bacterial vaccines in poultry has grown, and they are now widely recognized as an important tool for controlling bacterial infections in poultry. The early history of bacterial vaccines in poultry was marked by a focus on the development of killed or inactivated bacterial vaccines. These vaccines were made by growing a specific pathogen in the laboratory and then killing it with chemicals or heat to prevent it from causing disease. The killed bacteria were then used to produce the vaccine, which was administered to poultry to stimulate the production of antibodies that would protect against future infections with the same pathogen. Despite the initial success of these vaccines, they were limited by their inability to provide long-lasting protection against bacterial infections. The effectiveness of killed vaccines would decrease over time, requiring frequent booster shots to maintain protection. This led to a shift in focus towards the development of live, attenuated vaccines, which could provide longer-lasting protection against bacterial infections. Another important development in the history of bacterial vaccines in poultry was the introduction of recombinant vaccines. Recombinant vaccines are made by using genetic engineering techniques to produce a vaccine that contains a specific, targeted piece of the pathogen’s genetic material. These vaccines are highly specific, allowing them to target only the pathogen of concern without affecting other bacteria in the gut microbiome. This makes them a more sustainable alternative to antibiotics, as they do not contribute to the development of antibiotic-resistant bacteria. In recent years, there has been a growing interest in the development of bacteriophage-based vaccines. Bacteriophages are viruses that specifically target bacteria, and they have been shown to be highly effective at controlling bacterial infections in poultry. Bacteriophage-based vaccines work by introducing bacteriophages into the poultry, which then infect and destroy specific bacterial pathogens, reducing the need for antibiotics [7,28,29,30,31].

## 6. Properties of a Comparative Study of Different Bacterial Vaccines for Poultry

A comparative study of different bacterial vaccines for poultry involves evaluating and comparing the efficacy, safety, and cost-effectiveness of different bacterial vaccines available for poultry. This type of study is important because it helps to determine the best vaccine for a particular use based on the specific needs of the poultry industry.

There are several factors that can be considered in a comparative study of bacterial vaccines [32,33,34,35,36], including the following:Efficacy: This refers to the ability of the vaccine to effectively prevent or control the target bacterial infection. The efficacy of a vaccine can be determined through field trials, laboratory experiments, and/or observational studies.Safety: The safety of a vaccine is determined by evaluating the potential adverse effects associated with its use, such as local or systemic reactions, toxicity, and interference with other vaccines.Cost-effectiveness: The cost-effectiveness of a vaccine is determined by evaluating the cost of the vaccine in relation to the benefits it provides, such as reducing the need for antibiotics, improving productivity, and reducing the risk of AMR.Administration: The ease of administration of a vaccine can also be considered, including the route of administration, dosage, and storage requirements.Durability: The durability of a vaccine refers to its ability to provide long-lasting protection against the target bacterial infection.Spectrum of activity: The spectrum of activity of a vaccine refers to the range of bacterial strains that the vaccine can protect against.

By evaluating these factors, a comparative study of different bacterial vaccines for poultry can provide valuable information on the relative strengths and weaknesses of different vaccines and help to determine the best vaccine for a particular use. It is important to note that the results of a comparative study may vary depending on the specific circumstances and conditions of the poultry industry, such as the type of poultry, the prevalence of bacterial infections, and the specific needs of the industry. As such, it is important to conduct ongoing studies to evaluate and compare the performance of different vaccines in different circumstances. A comparative study of different bacterial vaccines for poultry is an important tool for evaluating the efficacy, safety, and cost-effectiveness of different vaccines and for determining the best vaccine for a particular use. Ongoing studies are necessary to ensure that the best vaccine is selected for the specific needs of the poultry industry. Top of Form.

## 7. Potential Mechanisms of Action of Bacterial Vaccines in Poultry

Bacterial vaccines work by stimulating the poultry’s immune system to produce a response against specific pathogens. This provides protection against future infections with the same pathogen, reducing the need for antibiotics. There are several mechanisms of action of bacterial vaccines, each of which provides a different approach to controlling bacterial infections in poultry [37,38,39,40,41]:The first mechanism of action is the stimulation of the immune response. When a bird is vaccinated, its immune system produces a response against the vaccine, which includes the production of antibodies. These antibodies are specific to the pathogen targeted by the vaccine, and they provide protection against future infections with the same pathogen. This is known as active immunity, and it provides long-lasting protection against bacterial infections.The second mechanism of action is the competition for nutrients and attachment sites. Some bacterial vaccines work by introducing a benign, or “competitor” bacteria into the poultry’s gut. This competitor bacteria competes with pathogenic bacteria for nutrients and attachment sites in the gut, reducing their ability to colonize and cause disease. This is known as competitive exclusion, and it is a highly effective approach to controlling bacterial infections in poultry.The third mechanism of action of bacterial vaccine is a cell-mediated immune response. Bacterial vaccines also stimulate the production of T-cells, which are a type of white blood cell that can directly attack infected cells and help activate other cells in the immune system. T-cells can recognize specific bacterial antigens and release cytokines (small proteins that regulate the immune response) to activate other cells in the immune system.Another mechanism of action of bacterial vaccines is the stimulation of phagocytosis. Phagocytosis is the process by which immune cells called phagocytes engulf and destroy bacteria. Bacterial vaccines stimulate the production of phagocytes, such as macrophages and neutrophils, which can recognize and engulf bacteria.

Each of these mechanisms of action provides a different approach to controlling bacterial infections in poultry, and they have been shown to be highly effective in reducing the need for antibiotics. This is particularly important in the context of AMR, as the overuse of antibiotics in poultry production has contributed to the development of antibiotic-resistant bacteria.

## 8. Different Types of Bacterial Vaccines Used in Poultry and Their Efficacy

Bacterial vaccines are used to provide protection against specific bacterial infections in poultry. There are several different types of bacterial vaccines used in poultry. In this section, we will discuss the different types of bacterial vaccines used in poultry [7] as follows:Inactivated bacterial vaccines: Inactivated bacterial vaccines are made by killing the bacteria and then purifying and inactivating the vaccine. These vaccines stimulate the immune system to produce a response against the specific pathogen, providing long-lasting protection against future infections. Examples of inactivated bacterial vaccines used in poultry include vaccines against *Salmonella*, *Campylobacter*, and *E. coli*.Live attenuated bacterial vaccines: Live attenuated bacterial vaccines are made by attenuating or weakening the bacteria, so they are no longer harmful but still able to stimulate an immune response. These vaccines can provide long-lasting protection against future infections with the same pathogen. Examples of live attenuated bacterial vaccines used in poultry include vaccines against Newcastle disease, infectious bronchitis, and fowl pox.Subunit bacterial vaccines: Subunit bacterial vaccines are made by purifying and isolating specific proteins or antigens from the bacteria and then using these antigens to stimulate an immune response. These vaccines can be highly effective but may require multiple doses to provide long-lasting protection against future infections. Examples of subunit bacterial vaccines used in poultry include vaccines against *Salmonella*, *Campylobacter*, and *E. coli*.Recombinant bacterial vaccines: Recombinant bacterial vaccines are made by using genetic engineering techniques to introduce specific antigens into a harmless vector, such as a bacterium or yeast. These vaccines can provide highly effective protection against specific bacterial pathogens and may only require a single dose to provide long-lasting protection. Examples of recombinant bacterial vaccines used in poultry include vaccines against *Salmonella* and *Campylobacter*.

The efficacy of bacterial vaccines in poultry can vary depending on several factors, including the type of vaccine, the specific pathogen being targeted, the age and health status of the poultry, and the management practices used in the poultry production system [7]. In general, bacterial vaccines have been shown to be highly effective in controlling bacterial infections in poultry, reducing the need for antibiotics, and improving animal performance. However, it is important to note that bacterial vaccines are not a panacea and cannot provide protection against all bacterial pathogens. In addition, the efficacy of bacterial vaccines can be reduced by factors such as poor storage and handling, incorrect administration, or the presence of other diseases or stressors in poultry.

## 9. Commercially Available Vaccines for Different Common Poultry Bacterial Diseases

Commercially available bacterial vaccines for poultry are widely used to control and prevent bacterial infections in poultry farms. These vaccines are generally produced from killed or attenuated bacterial cells or their toxins, and they stimulate the bird’s immune system to produce an immune response that can protect against bacterial infections. Below are some commonly used bacterial vaccines for poultry [27,42]:*Salmonella* vaccine: This vaccine is used to prevent *Salmonella* infections in poultry. It is composed of killed bacteria or a live attenuated strain that has been modified to reduce its virulence. Vaccinating birds against a particular serovar that is specific to their host, such as *Salmonella gallinarum*, results in the development of a robust and targeted immune response. The vaccine can be administered through drinking water or injection. The largest selection of available vaccines is designed to target serovar Enteritidis and Typhimurium. These vaccines are typically given through subcutaneous injection when birds are between 10 to 14 weeks old, with two separate doses administered 4 to 6 weeks apart.Infectious coryza vaccine: This vaccine is used to protect against infectious coryza (caused by *Haemophilus paragallinarum* or *Avibacterium paragallinarum*), a bacterial respiratory disease that affects poultry. The vaccine is typically produced from killed or inactivated bacteria. In the United States and other nations, there are commercial bacterins that typically consist of all serovars of the bacterium. Certain vaccines made by large manufacturers are marketed globally and contain the most common bacterial strains. Nonetheless, there are worries that such vaccines may not protect against locally prevalent variants. These types of vaccines are normally administered through drinking water or injection.Avian *E. coli* vaccine: This vaccine is used to prevent *E. coli* infections in poultry. It is commonly produced from live attenuated bacteria and administered through drinking water or injection. However, currently, in the United States, there exists only a live attenuated vaccine option. This vaccine features a mutant strain with an *aroA* deletion. The administration of antibiotics is not allowed when using this vaccine.*Pasteurella multocida* vaccine: *Pasteurella multocida* vaccines are available in different forms, including bacterins combined with aluminum hydroxide or oil emulsions, or with weakened live organisms. Multivalent *P. multocida* vaccines usually have serotypes 1, 3, and 4, which are the most common. Inactivated vaccines are typically administered through injection, while attenuated live vaccines (using M9 or PM-1 strains) can be given through the wing web or drinking water. It takes about two weeks for immunity to develop after vaccination.Avian mycoplasma vaccine: This vaccine is used to prevent *Mycoplasma gallisepticum* and *Mycoplasma synoviae* infections in poultry. The vaccine is typically produced from killed or attenuated bacteria. Live MG vaccines are available in several types, such as the mild F strain, the safer avirulent ts-11 or 6/85 strains, etc. The F strain can be given through intranasal or eye drop methods, while the ts-11 strain is administered through eye drops and the 6/85 strain through fine spray. The use of these attenuated vaccines is considered controlled exposure, which means they cause only mild infection at an age when it is less damaging. The vaccination of pullets is generally employed between 12 to 16 weeks of age, and one dose is enough to make them permanent carriers. Moreover, a live MS vaccine with the MS-H strain is given by eye drop.

In addition to these vaccines, there are other commercially available bacterial vaccines for poultry, such as those used to prevent *Clostridium perfringens* infections. The summary of commercially available bacterial vaccines against various common bacterial diseases are documented in Table 2.

## 10. Advantages and Disadvantages of Using Bacterial Vaccines in Poultry

Bacterial vaccines have become an important tool for controlling bacterial infections in poultry, and for reducing the need for antibiotics. However, like all interventions, bacterial vaccines have both advantages and disadvantages that need to be considered when deciding whether to use them in a poultry production system.

Potential advantages of bacterial vaccines in poultry are listed in the following [47,48,49,50,51]:Reduced need for antibiotics: The primary advantage of bacterial vaccines is that they reduce the need for antibiotics. By stimulating the poultry’s immune system to produce a response against specific pathogens, bacterial vaccines provide protection against future infections with the same pathogen, reducing the need for antibiotics. This is particularly important in the context of AMR, as the overuse of antibiotics in poultry production has contributed to the development of antibiotic-resistant bacteria.Increased immunity: Bacterial vaccines can also provide increased immunity against specific bacterial pathogens. By stimulating the poultry’s immune system to produce a response against the pathogen, bacterial vaccines provide long-lasting protection against future infections with the same pathogen. This can improve the health and welfare of the poultry, as well as reduce the need for antibiotics.Improved animal performance: Improved immunity against bacterial infections can also result in improved animal performance. Poultry that are protected against bacterial infections are less likely to experience disease, reducing the need for antibiotics and improving overall health. This can result in improved feed conversion, weight gain, and egg production.Reduced spread of antibiotic-resistant bacteria: Bacterial vaccines can also help to reduce the spread of antibiotic-resistant bacteria. By controlling bacterial infections in poultry, bacterial vaccines reduce the need for antibiotics, and, in turn, reduce the exposure of bacteria to antibiotics. This reduces the selection pressure for antibiotic-resistant bacteria, reducing their spread to other poultry and to the environment.Reduced risk of transmission to humans: Bacterial vaccines can also help to reduce the risk of transmission of bacterial infections from poultry to humans. By controlling bacterial infections in poultry, bacterial vaccines reduce the risk of bacterial pathogens being spread to humans through the food supply, reducing the risk of human infections.

Potential disadvantages of bacterial vaccines in poultry are listed in the following [8,30,52,53,54]:Cost: One of the main disadvantages of bacterial vaccines is their cost. Bacterial vaccines can be more expensive than antibiotics, particularly in large-scale poultry production systems. This can make them less accessible to some producers, particularly in developing countries.Ineffectiveness against some pathogens: Bacterial vaccines may not be effective against all bacterial pathogens, and some may be better suited to certain types of infections than others. This means that producers need to carefully select the most appropriate bacterial vaccine for their needs.Time to efficacy: Bacterial vaccines may take several weeks to become effective, during which time the poultry may still be susceptible to infection. This can result in a period of increased risk and may require the use of antibiotics in the meantime.Limited availability: Bacterial vaccines may not be widely available in all countries, particularly in developing countries. This can limit their use in some regions, and producers may need to import vaccines from other countries, which can be costly and logistically challenging.

## 11. The Role of Bacterial Vaccines in Reducing the Use of Antibiotics and the Development of Antimicrobial Resistance in Poultry Top of Form

The use of antibiotics in poultry production has been a topic of concern for several decades due to the growing problem of AMR [55]. Bacterial vaccines have emerged as a promising alternative to the use of antibiotics in poultry production. Bacterial vaccines work by stimulating the immune system of poultry to produce a response against specific bacterial pathogens, providing long-lasting protection against future infections. By reducing the incidence of bacterial infections in poultry, bacterial vaccines can reduce the need for antibiotics, helping to slow the development of AMR (Figure 2).

The role of bacterial vaccines in reducing the use of antibiotics in poultry can be seen in several different ways [9,48,56,57,58,59,60]: (a) Bacterial vaccines can help to prevent bacterial infections in the first place. By stimulating the immune system of poultry, bacterial vaccines can provide long-lasting protection against specific bacterial pathogens, reducing the need for antibiotics to treat bacterial infections. This can lead to a significant reduction in the use of antibiotics, helping to slow the development of AMR. (b) Bacterial vaccines can help to reduce the severity of bacterial infections. By stimulating the immune system of poultry, bacterial vaccines can help to reduce the severity of bacterial infections, reducing the need for antibiotics to treat the infections. This is especially important in the context of AMR, as reducing the severity of bacterial infections can reduce the need for antibiotics and help to slow the development of resistance. (c) Bacterial vaccines can help to improve the overall health and performance of poultry. By reducing the incidence and severity of bacterial infections, bacterial vaccines can help to improve the overall health and performance of poultry, reducing the need for antibiotics to treat bacterial infections and improving the efficiency of poultry production. This can lead to improved economic outcomes for poultry producers, as well as better health outcomes for the poultry themselves.

## 12. Comparison of Bacterial Vaccines with Other Alternative Strategies for Combating Antimicrobial Resistance in Poultry

Bacterial vaccines are one of several alternative strategies available for combating AMR in poultry. Other alternative strategies include the use of probiotics, prebiotics, bacteriophages, and dietary modifications. In this section, we will compare bacterial vaccines with these alternative strategies to understand their relative merits and drawbacks [61,62,63,64,65,66,67,68,69,70]:Probiotics are live microorganisms that, when ingested in adequate amounts, have a beneficial effect on the host. They can include beneficial bacteria, such as *Lactobacillus* and *Bifidobacterium*, as well as yeast and other microorganisms. In poultry, probiotics can be used to improve gut health and reduce the colonization of pathogenic bacteria. However, probiotics are limited in their ability to target specific pathogens and may have limited efficacy in controlling AMR.Prebiotics, on the other hand, are non-digestible food ingredients that promote the growth of beneficial bacteria in the gut. They can include substances such as fructooligosaccharides, inulin, and mannan-oligosaccharides. By promoting the growth of beneficial bacteria, prebiotics can reduce the colonization of pathogenic bacteria and limit the spread of AMR. However, prebiotics are limited in their ability to target specific pathogens and may have limited efficacy in controlling AMR.Bacteriophages are viruses that infect and kill bacteria. In poultry, bacteriophages can be used to control specific bacterial pathogens and limit the spread of AMR. However, the development and implementation of bacteriophage-based interventions can be challenging due to the limited availability of specific bacteriophages for different bacterial pathogens and the potential for the development of phage resistance.Dietary modifications, such as the inclusion of plant-based compounds, can also be used to combat AMR in poultry. For example, the inclusion of essential oils (e.g., substances such as thyme, cinnamon, and eucalyptus oil) and other plant-based compounds (e.g., garlic extract, turmeric extract, ginger extract, etc.) in the diet of poultry has been shown to have antimicrobial activity and reduce the colonization of pathogenic bacteria. However, the efficacy of dietary modifications in controlling AMR can be limited and may not be as effective as bacterial vaccines.Nanoparticles have shown promising potential as antimicrobial agents due to their unique physicochemical properties, which can enhance their efficacy against a broad range of microorganisms. However, the use of nanoparticles as antimicrobial agents also has some limitations that need to be addressed. One of the main concerns is the potential toxicity of nanoparticles to poultry, humans, and the environment. Moreover, the mechanisms of action of nanoparticles as antimicrobial agents are not yet fully understood, which can make it challenging to optimize their efficacy and safety. These limitations highlight the need for further research to better understand the potential benefits and risks of using nanoparticles as antimicrobial agents.In comparison, bacterial vaccines offer several advantages over these alternative strategies. Bacterial vaccines are specific to the targeted pathogen and provide a long-lasting immunity that can reduce the colonization of pathogenic bacteria and limit the spread of AMR. Additionally, bacterial vaccines do not have the potential for the development of resistance, as seen with the use of antibiotics and other antimicrobial agents. However, it should be noted that bacterial vaccines are not a cure-all solution for AMR in poultry and may not be effective against all bacterial pathogens. The development and implementation of bacterial vaccines can also be challenging and may require significant investments in research and development.Bacterial vaccines offer several advantages over alternative strategies for combating AMR in poultry. However, the relative efficacy of bacterial vaccines in comparison to alternative strategies will depend on the specific bacterial pathogen and the underlying conditions in the poultry production environment. Further research is needed to fully understand the potential of bacterial vaccines and their relative efficacy in comparison to other alternative strategies.

## 13. Case Studies on Bacterial Vaccines in Poultry Farming

Case studies are an important tool for evaluating the real-world impact of bacterial vaccines in poultry farming. By examining specific examples of the use of bacterial vaccines, case studies can provide valuable information on the effectiveness, safety, and cost-effectiveness of these vaccines in a real-world setting. Case studies typically involve collecting data from a specific poultry farm or group of farms, evaluating the impact of a bacterial vaccine on the health and productivity of the poultry, and comparing the results to those obtained from a control group of poultry that was not vaccinated.

Some common measures used in case studies on bacterial vaccines in poultry farming include the following [71,72]:Disease incidence: This refers to the number of poultry that develop the target bacterial infection after vaccination.Mortality rate: This refers to the number of poultry that die as a result of the target bacterial infection.Productivity: This refers to measures of the poultry’s performance, such as weight gain, feed efficiency, and egg production.Antibiotic use: This refers to the use of antibiotics on the vaccinated poultry and the control group and can provide valuable information on the impact of the vaccine on the need for antibiotics.Economic impact: This refers to the costs associated with the use of the vaccine, including the cost of the vaccine itself, the cost of administering the vaccine, and the cost of any associated treatments.

By analyzing these measures, case studies can provide valuable information on the real-world impact of bacterial vaccines in poultry farming. They can also provide valuable insights into the benefits and limitations of using these vaccines and help to identify best practices for their use. It is important to note that the results of a case study may vary depending on the specific circumstances and conditions of the poultry farm, such as the type of poultry, the prevalence of bacterial infections, and the specific needs of the farm. As such, it is important to conduct ongoing case studies to evaluate the impact of bacterial vaccines in different circumstances and to identify best practices for their use. Case studies are an important tool for evaluating the real-world impact of bacterial vaccines in poultry farming. By providing valuable information on the effectiveness, safety, and cost-effectiveness of these vaccines, case studies can help to improve the use of bacterial vaccines in poultry farming and reduce the impact of AMR.

## 14. A Review of Current Research Studies on the Efficacy of Bacterial Vaccines in Poultry

In recent years, there has been a growing body of research on the efficacy of bacterial vaccines in reducing the incidence of bacterial infections and the associated use of antibiotics in poultry production. These studies have sought to evaluate the impact of bacterial vaccines on the overall health and performance of poultry, as well as their ability to reduce the risk of AMR. This section provides a review of some of the key research studies on the efficacy of bacterial vaccines in poultry and their impact on AMR.

### 14.1. Studies on the Efficacy of Bacterial Vaccines in Poultry

One of the most promising areas of research has been the evaluation of bacterial vaccines for the prevention of infections caused by *Salmonella* and *E. coli* in poultry.

For example, Dórea et al. [73] conducted research using a live-attenuated *Salmonella* vaccine on commercial poultry farms. The research showed that vaccinated chickens had a decreased prevalence of *Salmonella* compared to non-vaccinated hens in both the reproductive tracts (14.22% compared to 51.7%; *p* < 0.001) and the ceca (38.3% versus 64.2%; *p* < 0.001). Another study by Berghaus et al. [74] reported that the use of a killed *Salmonella* vaccine reduced the incidence of *Salmonella* infection in broiler chicken flocks by more than 60%. Moreover, several previous studies recorded that the use of the *Salmonella* vaccine reduced the occurrence of *Salmonella* infections in poultry [33,75,76,77,78,79,80].

Lozica et al. [81] reported that an autonomous *E. coli* vaccine significantly reduced morbidity and mortality and increased egg production in poultry. Several studies showed that an *E. coli* vaccine, Poulvac® (Zoetis, Parsippany, NJ, USA) effectively worked against the avian pathogenic *E. coli* in poultry [82,83,84,85,86,87].

Another area of research has been the evaluation of bacterial vaccines for the prevention of *Campylobacter* infections in poultry. For example, a study by Clark et al. [88] reported that a vaccine against *Campylobacter* significantly reduced the colonization of *Campylobacter* infections in chickens. Similar results were recorded in the previous studies conducted for the reduction of *Campylobacter* infection in poultry [89,90,91,92,93,94,95,96,97].

Moreover, the use of bacterial vaccines against *M. gallisepticum* and *P. multocida* reduced their bacterial loads and improved the immunity in poultry [98,99,100,101,102,103,104,105,106].

### 14.2. Studies on the Impact of Bacterial Vaccines on Antimicrobial Resistance

In addition to their efficacy in reducing the incidence of bacterial infections, bacterial vaccines have also been shown to have a positive impact on the risk of AMR. For example, a study by Śmiałek et al. [107] found that the vaccination of chickens against colibacillosis using a gene-deleted live vaccine was the most effective way to increase the antibiotic susceptibility of *E. coli* isolates found in the field. Moreover, in another study [51], they also showed that the use of live attenuated vaccines reduced the use of antibiotics to improve the situation of multidrug-resistant *E. coli* in broilers. Another area of research has been the evaluation of the impact of bacterial vaccines on the development of AMR in bacteria that are commonly associated with poultry.

## 15. The global Market for Bacterial Vaccines in Poultry and Current Trends

The global market for bacterial vaccines in poultry is growing at a significant pace as the need for sustainable and cost-effective solutions to combat AMR increases. Over the course of the analysis period, it is anticipated that the market for poultry vaccines will demonstrate a noteworthy growth rate of 6% [108]. The growth of the global market for bacterial vaccines in poultry is driven by a number of factors, including the following [108,109]:Growing concern over AMR: The rise of AMR is a growing concern worldwide, and bacterial vaccines are seen as an effective solution for controlling AMR in poultry.Increased demand for meat: With a growing global population, the demand for meat is increasing, leading to an increase in the number of poultry operations. This has resulted in a growing demand for bacterial vaccines to control the spread of AMR in poultry.Government initiatives: Governments worldwide are taking steps to reduce the use of antibiotics in animal production and promoting the use of alternatives, including bacterial vaccines.Research and development: The bacterial vaccines market in poultry is supported by ongoing research and development initiatives aimed at improving the efficacy of bacterial vaccines and developing new vaccines to target specific bacterial pathogens.Increase in poultry production: The growth of the global poultry industry is a significant factor driving the growth of the bacterial vaccines market in poultry. As the number of poultry operations increases, so does the demand for bacterial vaccines to control the spread of AMR.

Currently, the Asia-Pacific region is the largest market for vaccines in poultry, followed by Europe and North America. The growth of the bacterial vaccines market in the Asia-Pacific region is driven by the increasing demand for meat in countries such as China, India, and Indonesia [109]. The global market for bacterial vaccines in poultry is growing at a significant pace as the need for sustainable and cost-effective solutions to combat AMR increases. With ongoing research and development, increasing demand for meat, and government initiatives aimed at reducing the use of antibiotics in animal production, the bacterial vaccines market in poultry is expected to continue to grow in the coming years.

## 16. The Future Potential of Bacterial Vaccines in Poultry and their Potential Impact on the Poultry Industry

In recent years, the development and use of bacterial vaccines in poultry has gained increasing attention as a means of reducing the use of antibiotics and the risk of AMR. Despite the promising results of current research studies, there is still much to be learned about the potential of bacterial vaccines in poultry and their impact on the poultry industry. In this section, we will discuss some of the key factors that are likely to influence the future potential of bacterial vaccines in poultry and their impact on the poultry industry.

### 16.1. The Development of New and Improved Bacterial Vaccines

One of the key factors that is likely to influence the future potential of bacterial vaccines in poultry is the development of new and improved vaccines. With advances in vaccine technology and a growing understanding of the mechanisms of bacterial infections in poultry, it is likely that new vaccines will emerge that are more effective and easier to use. For example, research is currently underway to develop vaccines that can be delivered orally, which would greatly simplify the vaccination process for poultry producers.

### 16.2. The Role of Government and Industry in the Promotion of Bacterial Vaccines

Another important factor that is likely to influence the future potential of bacterial vaccines in poultry is the role of the government and industry in promoting their use. In many countries, there are already efforts underway to encourage the use of bacterial vaccines in poultry as a means of reducing the use of antibiotics and the risk of AMR. For example, many governments are providing funding for research into the development of new and improved bacterial vaccines and are also implementing regulations that require poultry producers to use vaccines as a condition of licensing.

### 16.3. The Impact of Bacterial Vaccines on the Poultry Industry

The use of bacterial vaccines in poultry is likely to have a significant impact on the poultry industry in the coming years. By reducing the incidence of bacterial infections and the associated use of antibiotics, bacterial vaccines are likely to improve the overall health and performance of poultry, as well as reduce the risk of AMR. This, in turn, is likely to increase consumer confidence in the safety and quality of poultry products and help to protect the reputation of the poultry industry.

However, the implementation of bacterial vaccines in poultry production is likely to present some challenges for poultry producers. For example, the cost of vaccines and the necessary equipment for their administration may be a barrier for some producers, particularly those in developing countries. In addition, the lack of infrastructure and trained personnel to administer vaccines in some countries may also pose a challenge to the implementation of bacterial vaccines in poultry production. Top of Form.

## 17. The impact of Bacterial Vaccines on the Welfare of Poultry and Public Health Top of Form

The impact of bacterial vaccines on the welfare of poultry and public health concerns is an important topic of discussion. Bacterial vaccines have been shown to improve the health and welfare of poultry by reducing the incidence of bacterial infections and reducing the need for antibiotics. By preventing bacterial infections, bacterial vaccines can help to reduce the suffering and mortality of poultry, improve their overall health and well-being, and increase the efficiency of poultry production [7]. In terms of public health concerns, bacterial vaccines are generally considered safe for use in poultry and are unlikely to pose a risk to human health. Unlike antibiotics, which can contribute to the development of antibiotic-resistant bacteria, bacterial vaccines do not select for antibiotic-resistant bacteria. Instead, they stimulate the immune system of poultry to provide protection against specific bacterial infections, reducing the need for antibiotics and reducing the risk of the development of antibiotic-resistant bacteria [110]. However, as with any veterinary medicine, there is always a risk of adverse reactions to bacterial vaccines in poultry. Some poultry may experience adverse reactions, such as swelling or redness at the site of injection, but these are generally mild and temporary. In rare cases, some poultry may experience more serious adverse reactions, such as anaphylaxis, but these are extremely rare. The impact of bacterial vaccines on the welfare of poultry and public health concerns is generally positive. Bacterial vaccines have been shown to improve the health and welfare of poultry, reduce the incidence of bacterial infections, and reduce the need for antibiotics [6]. They are considered safe for use in poultry and unlikely to pose a risk to human health. However, as with any veterinary medicine, there is always a risk of adverse reactions to bacterial vaccines in poultry, and it is important to monitor the welfare of poultry and assess any potential risks to human health. Top of Form.

## 18. The Economic Benefits of Using Bacterial Vaccines in Poultry and Cost-Effectiveness Analysis

The economic benefits of using bacterial vaccines in poultry and cost-effectiveness analysis are important considerations for the poultry industry. Bacterial vaccines can provide numerous economic benefits, including reduced mortality and morbidity rates, increased feed conversion efficiency, improved egg production, and reduced veterinary costs. By reducing the incidence of bacterial infections and reducing the need for antibiotics, bacterial vaccines can help to increase the efficiency of poultry production and reduce the overall cost of production. In terms of cost-effectiveness analysis, the use of bacterial vaccines in poultry can be seen as a cost-effective alternative to the use of antibiotics. Unlike antibiotics, which can contribute to the development of antibiotic-resistant bacteria and have negative impacts on the environment, bacterial vaccines do not select for antibiotic-resistant bacteria and are considered environmentally friendly. Additionally, the cost of bacterial vaccines is often lower than the cost of antibiotics, making them a cost-effective alternative for controlling bacterial infections in poultry. However, it is important to consider the cost of developing and implementing a vaccination program, as well as the cost of the vaccines themselves. While bacterial vaccines can provide numerous economic benefits, they may not always be cost-effective in all situations, and it is important to perform a detailed cost-effectiveness analysis to determine the most cost-effective strategy for combating AMR in poultry. The economic benefits of using bacterial vaccines in poultry and cost-effectiveness analysis are important considerations for the poultry industry. Bacterial vaccines can provide numerous economic benefits, including reduced mortality and morbidity rates, increased feed conversion efficiency, improved egg production, and reduced veterinary costs. While the cost of developing and implementing a vaccination program and the cost of the vaccines themselves must be considered, bacterial vaccines can be seen as a cost-effective alternative to the use of antibiotics for controlling bacterial infections in poultry [9,111].

## 19. The Impact of Bacterial Vaccines on the Environment and Sustainability of the Poultry Industry

The use of bacterial vaccines in the poultry industry has the potential to significantly impact the environment and sustainability of this sector. Bacterial vaccines provide a targeted and effective method of controlling bacterial infections in poultry, reducing the need for antibiotics and the use of chemical treatments. By reducing the use of antibiotics, bacterial vaccines can help to reduce the development of AMR and preserve the effectiveness of antibiotics for future use. In addition, the reduction in the use of antibiotics and chemical treatments can have positive impacts on the environment. Antibiotics and other chemical treatments can enter the environment through poultry manure, which can have negative effects on soil and water quality. Bacterial vaccines can help to reduce the release of these substances into the environment, improving environmental sustainability. Furthermore, the reduction in bacterial infections in poultry through the use of bacterial vaccines can result in healthier poultry, which can have a positive impact on the poultry industry’s sustainability. Healthier poultry are more efficient, with increased growth rates, improved feed conversion, and lower mortality rates. This can increase the overall profitability of the poultry industry, helping to ensure its sustainability in the long term. However, there are also some potentially negative impacts of bacterial vaccines on the environment that must be considered. For example, the production and disposal of vaccines can result in the release of pollutants into the environment, which may have negative impacts on soil and water quality. Additionally, the production of vaccines requires energy and resources and may result in greenhouse gas emissions, contributing to climate change. The impact of bacterial vaccines on the environment and sustainability of the poultry industry is complex and multi-faceted. On the one hand, bacterial vaccines can help to reduce the use of antibiotics and chemical treatments, improving environmental sustainability. On the other hand, there are potential negative impacts associated with the production and disposal of vaccines that must also be considered. Overall, the use of bacterial vaccines in the poultry industry has the potential to provide positive benefits for both animal health and environmental sustainability but must be carefully managed to minimize negative impacts [112,113,114].

## 20. The Regulatory Landscape for Bacterial Vaccines in Poultry and Challenges to Their Widespread Adoption

The regulatory landscape for bacterial vaccines in poultry is complex and varies from country to country. In many countries, bacterial vaccines are subject to strict regulatory approval processes before they can be marketed and used. These processes are aimed at ensuring the safety and efficacy of bacterial vaccines and protecting the health of consumers and the environment. In the European Union, bacterial vaccines are subject to the same regulations as other veterinary medicines and must be authorized by the European Medicines Agency (EMA) before they can be used. In the United States, bacterial vaccines are regulated by the Food and Drug Administration (FDA) and must undergo a rigorous approval process before they can be marketed and used [115]. One of the challenges to the widespread adoption of bacterial vaccines in poultry is the lack of standardized regulatory frameworks in many countries. In some countries, the regulatory approval process for bacterial vaccines is more lenient than in others, leading to differences in the quality and efficacy of bacterial vaccines available in different markets. This makes it difficult for poultry producers to make informed decisions about which bacterial vaccines to use and raises concerns about the safety and efficacy of some vaccines. Another challenge to the widespread adoption of bacterial vaccines in poultry is the lack of awareness and understanding of the benefits of bacterial vaccines among poultry producers and consumers. Some poultry producers are hesitant to use bacterial vaccines because of the perceived complexity of the regulatory approval process and the lack of knowledge about the benefits of bacterial vaccines. Consumers are also concerned about the use of vaccines in animal production, raising questions about the safety of food produced with the use of vaccines. The regulatory landscape for bacterial vaccines in poultry is complex and varies from country to country. One of the challenges to the widespread adoption of bacterial vaccines is the lack of standardized regulatory frameworks and the lack of awareness and understanding of the benefits of bacterial vaccines among poultry producers and consumers. Addressing these challenges will be crucial to promoting the widespread adoption of bacterial vaccines and reducing the use of antibiotics in poultry production [115].

## 21. Challenges in the Development and Implementation of Bacterial Vaccines in Poultry

While bacterial vaccines have emerged as a promising alternative to the use of antibiotics in poultry production, there are several challenges that must be overcome in order to fully realize their potential in reducing the use of antibiotics. These challenges can be grouped into several different categories, including scientific, technical, and economic challenges.

### 21.1. Scientific Challenges

One of the major scientific challenges in the development of bacterial vaccines for poultry is the need to identify and understand the key factors that contribute to bacterial resistance. This requires a detailed understanding of the mechanisms of resistance, as well as the ability to identify and target key virulence factors that are essential for the survival and replication of bacteria. Another major scientific challenge is the need to develop bacterial vaccines that are effective against a wide range of bacterial pathogens. This requires a deep understanding of the biology and genetics of bacterial pathogens, as well as the ability to design and synthesize vaccines that are effective against multiple bacterial pathogens [116,117]. This is particularly important in the context of poultry production, as poultry are commonly exposed to a wide range of bacterial pathogens, including *E. coli*, *Salmonella*, and *Campylobacter*.

### 21.2. Technical Challenges

Technical challenges can also pose a barrier to the successful implementation of bacterial vaccines in poultry. One of the major technical challenges is the need to ensure that bacterial vaccines are stored and handled properly in order to maintain their efficacy. This requires careful attention to storage and handling protocols, as well as the ability to transport vaccines to remote locations where poultry production is taking place. Another major technical challenge is the need to ensure that bacterial vaccines are administered correctly in order to achieve maximum efficacy. This requires a deep understanding of the biology of poultry, as well as the ability to develop and implement effective administration protocols that are easy to follow and consistently achieve high levels of efficacy [116,118].

### 21.3. Economic Challenges

Economic challenges are another major barrier to the successful implementation of bacterial vaccines in poultry. One of the major economic challenges is the cost of developing and producing bacterial vaccines. This requires significant investments in research and development, as well as the ability to scale up production in order to meet the growing demand for bacterial vaccines. Another major economic challenge is the cost of implementing bacterial vaccine programs in poultry production. This requires investments in training, equipment, and infrastructure, as well as the ability to effectively market and promote bacterial vaccines to poultry producers [118,119,120].

## 22. The Role of Veterinary Clinics and Poultry Farmers in Promoting the Use of Bacterial Vaccines

The role of veterinary clinics and poultry farmers in promoting the use of bacterial vaccines is crucial in ensuring their success and widespread adoption. Veterinary clinics play a key role in educating poultry farmers on the benefits of using bacterial vaccines and providing guidance on their use and administration. They can also help to monitor and evaluate the efficacy of the vaccines, ensuring that they are providing the desired protection against bacterial infections. Poultry farmers also play a critical role in promoting the use of bacterial vaccines. By using bacterial vaccines, farmers can reduce the incidence of bacterial infections in their flocks and reduce the need for antibiotics. This not only improves the health and welfare of their poultry but also increases their overall efficiency and profitability. In order to promote the use of bacterial vaccines, veterinary clinics and poultry farmers must work together to ensure that the vaccines are properly stored, administered, and monitored. They must also work to overcome any barriers to their adoption, such as lack of knowledge or access to the vaccines, and address any concerns that poultry farmers may have about their use. The role of veterinary clinics and poultry farmers in promoting the use of bacterial vaccines is crucial to ensuring their success and widespread adoption. Both groups must work together to educate poultry farmers, monitor the efficacy of the vaccines, and address any barriers to their use. By promoting the use of bacterial vaccines, veterinary clinics and poultry farmers can help to reduce the incidence of bacterial infections and the need for antibiotics in poultry, improving the health and welfare of their poultry and increasing their overall efficiency and profitability. Top of Form.

## 23. The Role of Technology and Innovation in the Development of New and More Effective Bacterial Vaccines for Poultry

The poultry industry is a vital sector of agriculture and provides food for billions of people globally. The health and well-being of poultry flocks are essential for the production of safe and high-quality food. One of the biggest challenges facing the poultry industry is the emergence and spread of antimicrobial-resistant bacteria. AMR is a growing global threat to public health and animal health, and it is essential to find new and innovative solutions to combat this problem. One such solution is the development and use of bacterial vaccines. The role of technology and innovation in the development of new and more effective bacterial vaccines for poultry cannot be overstated. Advancements in genetic engineering, molecular biology, and biotechnology have significantly impacted the design, production, and delivery of bacterial vaccines in recent years. For example, genetic engineering has enabled the development of genetically modified bacteria that are used as carriers for vaccine antigens, thereby providing improved protection against diseases. Innovation in vaccine delivery systems has also played a critical role in the development of more effective bacterial vaccines. For example, the development of newer delivery systems, such as in-ovo (in-egg) vaccines, has greatly improved the efficacy of bacterial vaccines by increasing the uptake of antigens by poultry. The in-ovo vaccine delivery system eliminates the need for multiple injections, reducing the stress on the poultry, and providing protection from the early stages of life. The use of adjuvants in bacterial vaccines has also significantly impacted their efficacy. Adjuvants are substances that are added to vaccines to enhance the immune response to antigens. The development of new adjuvants, such as liposomes and microparticles, has led to the development of vaccines that are more effective and longer lasting. Technology and innovation play a critical role in the development of new and more effective bacterial vaccines for poultry. The use of genetic engineering, molecular biology, biotechnology, and vaccine delivery systems has enabled the production of vaccines that provide improved protection against AMR bacteria. The continued investment in research and development of new technologies will be crucial in the fight against AMR and the development of effective and safe bacterial vaccines for poultry [116,121,122,123]. Top of Form.

## 24. The Role of Public–Private Partnerships in Advancing the Development and Implementation of Bacterial Vaccines in Poultry

The role of public–private partnerships (PPP) in advancing the development and implementation of bacterial vaccines in poultry is crucial. PPPs bring together the resources, expertise, and perspectives of both the public and private sectors to address complex problems and achieve common goals. In the case of bacterial vaccines in poultry, PPPs can help to accelerate the development and implementation of new and more effective vaccines, as well as promote their widespread adoption. There are several ways in which PPPs can contribute to the advancement of bacterial vaccines in poultry [111,124,125].

PPPs can help to provide the funding and resources needed for research and development of new vaccines. By pooling resources from both the public and private sectors, PPPs can ensure that there is sufficient funding for R&D activities, which is critical for the development of new and more effective vaccines.PPPs can help to facilitate collaboration between researchers, industry, and government. Through collaboration, the different stakeholders can share their expertise, knowledge, and resources, which can result in more effective solutions. For example, researchers can benefit from the real-world knowledge and experience of poultry farmers and industry experts, while the industry can benefit from the latest scientific advancements in vaccine development.PPPs can help to promote the widespread adoption of bacterial vaccines in poultry. Through joint efforts, public and private sectors can raise awareness about the benefits of using bacterial vaccines and encourage poultry farmers to adopt these vaccines. By working together, PPPs can help to overcome any barriers to the adoption of vaccines, such as the lack of information or lack of access to vaccines.

PPPs play an important role in advancing the development and implementation of bacterial vaccines in poultry. By bringing together the resources, expertise, and perspectives of both the public and private sectors, PPPs can help to ensure that new and more effective vaccines are developed and adopted. Through joint efforts, PPPs can help to promote the widespread use of bacterial vaccines, which is critical for combating AMR in poultry. Top of Form.

## 25. An Overview of Current Global Efforts to Combat Antimicrobial Resistance in Poultry

Current global efforts to combat AMR in poultry focus on several key areas, including improving disease management, reducing the use of antibiotics, and promoting the use of alternative strategies, such as bacterial vaccines. The World Health Organization (WHO) and the Food and Agriculture Organization (FAO) of the United Nations have both emphasized the importance of reducing the use of antibiotics in poultry production and the potential of bacterial vaccines to play a key role in achieving this goal [126]. Several countries have implemented national strategies to reduce the use of antibiotics in poultry production which has a well-established national program promoting the use of bacterial vaccines. In addition, the European Union has adopted a One Health approach to combat AMR, recognizing the interconnection between human, animal, and environmental health. International organizations, such as the World Organization for Animal Health (OIE), also play a key role in promoting the use of bacterial vaccines in poultry. The OIE has developed guidelines for the evaluation and quality control of bacterial vaccines and provides support for their implementation in member countries [127]. Private sector companies, including vaccine manufacturers and poultry producers, also have a role to play in advancing the use of bacterial vaccines in poultry. Vaccine manufacturers are investing in the development of new and more effective bacterial vaccines, while poultry producers are implementing these vaccines in their production practices. There is a growing recognition of the potential for bacterial vaccines to play a key role in reducing the use of antibiotics in poultry production and combating AMR. Global efforts to promote the use of bacterial vaccines in poultry, including national strategies, international organizations, and private sector companies, are critical for achieving this goal [128]. However, continued research and investment in the development and implementation of effective bacterial vaccines are necessary to fully realize their potential impact. Top of Form.

Top of Form.

## 26. Conclusions and Recommendations

Bacterial vaccines have shown promising results in combating AMR in poultry. Bacterial vaccines provide a specific and long-lasting immunity against targeted bacterial pathogens and have the potential to reduce the spread of AMR. However, the development and implementation of bacterial vaccines are not without challenges. Further research is needed to fully understand the mechanisms of action of bacterial vaccines, their relative efficacy in comparison to other alternative strategies, and their potential impact on the poultry industry.

In order to fully realize the potential of bacterial vaccines in combating AMR in poultry, it is recommended that the following research and implementation initiatives be pursued:Development of new and improved bacterial vaccines: Research should be conducted to develop new and improved bacterial vaccines that are specific to the targeted bacterial pathogens and that offer long-lasting immunity.Efficacy studies: Further research is needed to fully understand the efficacy of bacterial vaccines in controlling AMR in poultry. This includes large-scale trials to determine the effectiveness of bacterial vaccines in reducing the spread of AMR and the impact of bacterial vaccines on the poultry industry.Regulatory support: Regulators should provide support for the development and implementation of bacterial vaccines in poultry. This includes the establishment of a regulatory framework for the development and commercialization of bacterial vaccines and the provision of technical assistance for the implementation of bacterial vaccines in the field.Industry support: The poultry industry should provide support for the development and implementation of bacterial vaccines. This includes the provision of resources for research and development, the promotion of the use of bacterial vaccines in poultry production, and the establishment of incentives for the use of bacterial vaccines in the field.Education and outreach: Education and outreach efforts should be conducted to raise awareness of the importance of bacterial vaccines in combating AMR in poultry. This includes the development of educational materials for producers and consumers and the engagement of stakeholders in the poultry industry to promote the use of bacterial vaccines.

Overall, bacterial vaccines have the potential to play a significant role in combating AMR in poultry. However, it will require a concerted effort from researchers, regulators, the poultry industry, and the public to fully realize this potential.

## Figures and Tables

**Figure 1 vaccines-11-00616-f001:**
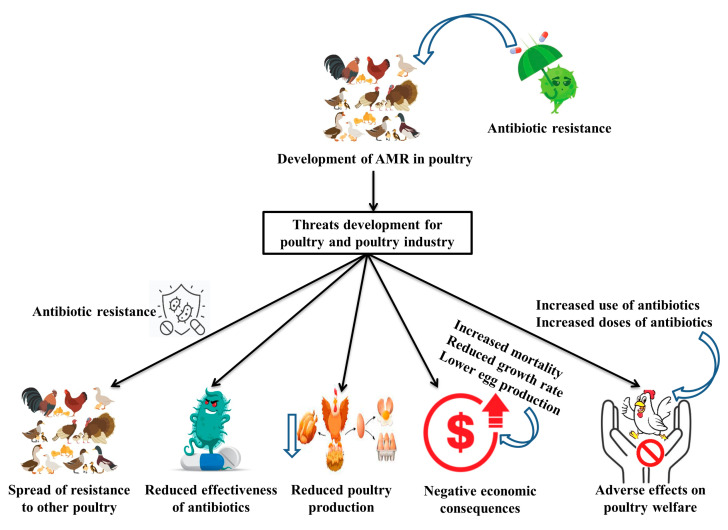
Possible threats of antimicrobial resistance on poultry and poultry production.

**Figure 2 vaccines-11-00616-f002:**
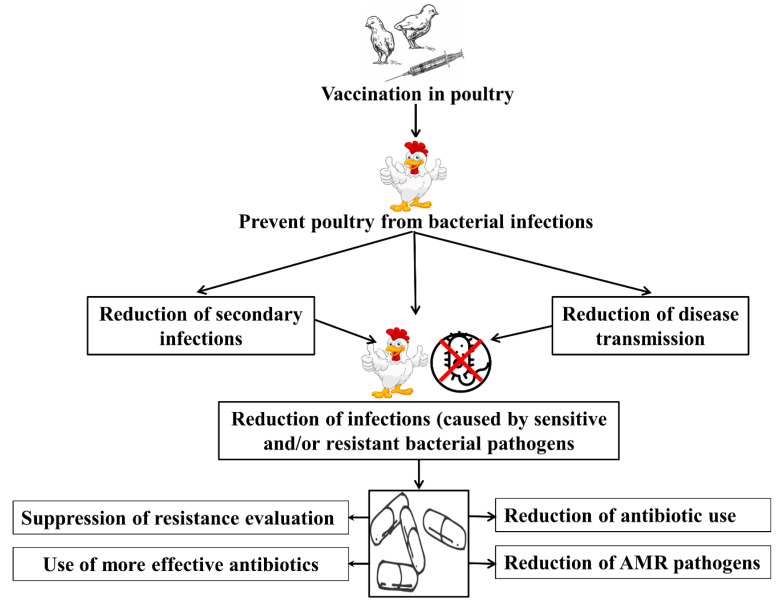
The role of bacterial vaccination in reducing the use of antibiotics and the occurrence of antimicrobial resistance in poultry.

**Table 1 vaccines-11-00616-t001:** List of bacterial diseases in poultry with their common clinical signs, mode of transmission, common treatments/control vaccines/drugs, and affected bird types [11,12,13,14].

Name of Diseases	Common Clinical Signs	Mode of Transmission	Common Treatment/Control Vaccine/Drugs	Affected Bird Types
Avian salmonellosis	Depression, poor growth, weakness, severe diarrhea, dehydration, and death.	Mainly egg transmission, others include mechanical transmission, carrier birds, contaminated premises, etc.	Treatment is mainly a salvage operation. However, antibiotics, e.g., furazolidone, gentamycin sulfate, and sulfa drugs can be used. Vaccines against local strain are used to control the disease.	Chickens, turkeys, ducks, pigeons, pheasants, and other game birds.
Avian colibacillosis	Ruffled feathers, fever, labored breathing, reduced appetite, poor growth, occasional coughing, rales, diarrhea, and sudden death.	Inhalation of the fecal contaminated dust.	Early treatment is recommended. Antibiotics such as tetracyclines, sulfas, ampicillin, and streptomycin maybe used to control some *E. coli.*	All types of poultry.
Avian Mycoplasmosis	Coughing, sneezing, respiratory rales, ocular and nasal discharge, decreased feed intake and egg production, increased mortality, poor hatchability, and, primarily in turkeys, swelling of the infraorbital sinus.	Vertically within some eggs (transovarian) from infected breeders to progeny, and horizontally via infectious aerosols and through contamination of feed, water, and the environment, and by human activity on fomites (shoes, equipment, etc.).	Can be treated with antibiotics to alleviate clinical symptoms. Tylosin, tilmicosin, and tiamulin are useful to reduce the mycoplasma load in the flock. However, antibiotic therapy cannot completely eliminate mycoplasma from the flock. Moreover, vaccines against local strain are used to control the disease.	Chickens and turkeys.
Pasteurellosis	Stupor, loss of appetite, rapid weight loss, lameness resulting from joint infection, swollen wattles, difficult breathing, watery yellowish or green diarrhea, cyanosis or darkening of the head and wattles, and sudden death.	Ingestion, mechanically by arthropod vectors or by inhalation.	Treatment is not practical, but when individual treatment is applicable, chlortetracycline, oxytetracycline, and sulfaquinoxaline can be used. Vaccines against a local strain are used to control the disease.	Chickens, turkeys, pheasants, pigeons, waterfowl, sparrows, and other free-flying birds.
Necrotic Enteritis	Severe depression, ruffled feathers, diarrhea, and sudden increased mortality.	Oral contact with the droppings from infected birds.	Bacitracin, penicillin, and lincomycin most often used.	Mainly broiler chickens. Layers and turkeys can also be affected.
Campylobacteriosis	Decreased egg production; death can occur rapidly.	Through a contaminated water source or through contact with feces.	Can be treated with antibiotics, e.g., azithromycin. Bacteriocin OR-7 treatment is also useful.	Broilers, layers, turkeys, ducks, and geese.
*Staphylococcus* infection	Affected chicks usually appear drowsy or droopy with the down being "puffed up". Diarrhea sometimes occurs. Mortality usually begins within 24 hours and peaks by 5-7 days.	Transmitted from unsanitary equipment in the hatchery to newly hatched birds having unhealed navels.	Staphylococcosis can be successfully treated with antibiotics, e.g., penicillin, erythromycin, lincomycin, and spectinomycin.	Chickens.
Infectious Coryza	Edematous swelling of the face around the eyes and wattles, nasal discharge and swollen sinuses.	By direct contact, airborne infection by dust or respiratory discharge droplets and drinking water contaminated by infective nasal exudate	A number of drugs (e.g., Sulfadimethoxine or sulfathiazole) are effective for treating the symptoms of the disease although the disease is never completely eliminated.	Chickens.
Chlamydiosis	Anorexia, ruffled feathers, apathy, drop in egg production, diarrhea, weight loss, ocular discharge, fever, and respiratory distress.	By the fecal-oral route or by inhalation.	Tetracyclines (chlortetracycline, oxytetracycline, doxycycline) are the antibiotics of choice.	Turkeys, ducks, and chickens.

**Table 2 vaccines-11-00616-t002:** List of commercially available bacterial vaccines against different common bacterial diseases in poultry production [27,43,44,45,46].

Name of Commercial Bacterial Vaccines	Vaccine Types	Name of Manufacturers, Country	Pathogens or Species	Name of Bacterial Diseases
AviPro^®^MEGAN^®^VAC 1	Live, attenuated vaccine Δ*cya*/Δ*crp* mutation	Elanco Animal Health, USA	*Salmonella* Typhimurium, *Salmonella* Enteritidis, and *Salmonella* Heidelberg	Avian salmonellosis
Vaxsafe^®^ST	Live, attenuated vaccine Δ*aroA* mutation	Bioproperties Pty Ltd., Australia	*Salmonella* Typhimurium	Avian salmonellosis
AviPro^®^*Salmonella* Vac E	Live, attenuated vaccine (Sm24/Rif12/Ssq strain)	Elanco Animal Health, USA	*Salmonella* Enteritidis	Avian salmonellosis
AviPro^®^Megan^®^Egg	Live, attenuated vaccine Δ*aroA* mutation	Elanco Animal Health, USA	*Salmonella* Enteritidis, *Salmonella* Typhimurium	Avian salmonellosis
Poulvac^®^ST	Live, attenuated vaccine Δ*aroA* mutation	Zoetis, USA	*Salmonella* Typhimurium	Avian salmonellosis
AviPro 109 SE4 Conc	Inactivated vaccine	Lohmann Animal Health, Germany	*Salmonella* Enteritidis	Avian salmonellosis
Gallivac^®^SE	Live, attenuated vaccine Δ*aroA* mutation	Merial Select, Italy	*Salmonella* Enteritidis	Avian salmonellosis
SALMUNE^®^	Live, attenuated vaccine	Ceva Animal Health, USA	*Salmonella* Typhimurium	Avian salmonellosis
Poulvac^®^*E. coli*	Live, attenuated vaccine Δ*aroA* mutation, O78 serotype	Zoetis, USA	*Escherichia coli*	Avian colibacillosis
Avipro 101 Coryza Gold	Inactivated (serotype A,B,C)	Lohmann Animal Health, Cuxhaven, Germany	*Haemophilus paragallinarum*	Infectious coryza
Coripravac-O	Killed [serotype A (strain 1753) + B (strain 1755) + C (strain 1756)	Hipra, Spain	*Avibacterium paragallinarum*	Infectious coryza
M-NINEVAX^®^-C	Live vaccine with mild avirulent M-9 strain	Merck, USA	*Pasteurella multocida*	Fowl cholera
Gallimune Cholera/Bio Chlolera	Inactivated (serotypes 1, 3 and 4.)	Merial Select, Italy	*Pasteurella multocida*	Fowl cholera
Multimune K5	Killed (serotypes 1, 3 & 4 + serotypes 3&4)	Biomune, USA	*Pasteurella multocida*	Fowl cholera
PM-ONEVAX^®^-C	Live vaccine with mild avirulent PM-1 strain	Merck, USA	*Pasteurella multocida*	Fowl cholera
MyVAC DP	Killed vaccine (serotype 1)	MVP Sdn. Bhd., Malaysia	*Pasteurella multocida*	Duck Pasteurellosis
MG TS-11	Live attenuated vaccine (TS-11 strain)	Merial Select, Italy	*Mycoplasma gallisepticum*	Avian mycoplasmosis
Gallimune MG/BioMyco/MG Vax	Killed vaccine (S6 strain)	Merial Select, Italy	*Mycoplasma gallisepticum*	Avian mycoplasmosis
AviPro^®^MG-F	Live attenuated vaccine (F strain)	Elanco Animal Health, USA	*Mycoplasma gallisepticum*	Avian mycoplasmosis
MG BacterinMS Bacterin	Bacterin (F strain)	Zoetis, USA	*Mycoplasma gallisepticum*, *Mycoplasma synoviae*	Avian mycoplasmosis
MYCOVAC-L^®^	Live vaccine (Intervet 6/85 strain)	Merck, USA	*Mycoplasma gallisepticum*	Avian mycoplasmosis
Myc-vac	Killed (NEV40 & NEV45 strain)	Fatro S.p.A, Italy	*Mycoplasma gallisepticum*	Avian mycoplasmosis
Poulvac^®^MycoF	Live attenuated vaccine (F strain)	Zoetis, USA	*Mycoplasma gallisepticum*	Avian mycoplasmosis

## Data Availability

Not applicable.

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
