# Peer review of "A Comprehensive Review on Bacterial Vaccines Combating Antimicrobial Resistance in Poultry"

_vaccines, 2023, doi:10.3390/vaccines11030616_

Round 1

Reviewer 1 Report

The review article entitle A Comprehensive Review on Bacterial Vaccines Combating Antimicrobial Resistance in Poultry is interesting and well written, I have few suggestions those should be include

·         Add a detail table list of bacterial disease with their common clinical signs, transmission, common treatment/control vaccine/drugs and specific affected bird type.

·         Add a brief detail of some phytotherapy treatment combating the bacterial infections in poultry.

·         Add brief detail about non-antibiotic microbial growth promoters

·         Add a graphical representation to threat of Antimicrobial Resistance in Poultry

·         Add a table or graph of Vaccine candidates in clinical development, categorized by pathogen and type.

·         Add overview of antibiotics contribute to antimicrobial resistance

Author Response

We would like to take this opportunity to thank the reviewers for their time and effort put into the review. The comments have helped to improve the revision significantly, and we humbly hope that the revision will be able to respond to the comments made by the reviewers adequately.

Comments and Suggestions for Authors

Reviewer 1:

The review article entitle A Comprehensive Review on Bacterial Vaccines Combating Antimicrobial Resistance in Poultry is interesting and well written, I have few suggestions those should be include

Response: Thank you very much for your valuable comments and suggestions to improve the manuscript.

  • Add a detail table list of bacterial disease with their common clinical signs, transmission, common treatment/control vaccine/drugs and specific affected bird type.

Response: Thank you very much for your valuable comments. We have added a detailed table (Table 1) for the common bacterial diseases in poultry.

  • Add a brief detail of some phytotherapy treatment combating the bacterial infections in poultry.

Response: Thank you very much for your valuable comments. The present review mainly focuses on the bacterial vaccines combating AMR in poultry. However, we already mentioned them very shortly in Section 12. We have now added more information about them as suggested.

  • Add brief detail about non-antibiotic microbial growth promoters

Response: Thank you very much for your valuable comments. The present review mainly focuses on the bacterial vaccines combating AMR in poultry. Nevertheless, we already mentioned them very shortly in Section 12. We have now added more information about them.

  • Add a graphical representation to threat of Antimicrobial Resistance in Poultry

Response: Thank you very much for your valuable comment. We have added a figure (Figure 1) in section 4 describing the possible threats of AMR on poultry and poultry production.

  • Add a table or graph of Vaccine candidates in clinical development, categorized by pathogen and type.

Response: Thank you very much for your valuable comment. We have added a table (Table 2) in section 9 that narrates the summary of commercially available vaccines against various bacterial diseases in poultry.

  • Add overview of antibiotics contribute to antimicrobial resistance

Response: Thank you very much for your valuable comment. We have added a new section (section 3) based on your suggestion.

Best regards,

Md. Tanvir Rahman (Corresponding Author) DVM, MSc (Canada), Ph.D. (UK), Postdoc (Germany)
Director, 
Professor Muhammed Hussain Central Laboratory, Professor of Microbiology,  Department of Microbiology and Hygiene,
Faculty of Veterinary Science, 
Bangladesh Agricultural University, Mymensingh-2202, Bangladesh. Phone. + 88-01913323307; Fax + 88-09161510 E.mail: [email protected]

Reviewer 2 Report

 This review describes the current knowledge of bacterial vaccines combating antimicrobial resistance in poultry. This is very well written review and will provide useful summarization of current status of the field. However, this review touched too many aspects of bacterial vaccines for poultry to provide any beneficial information for the expert readers. This review would provide very little useful information to the reader of the journal “Vaccines”. More detailed review of ‘the role of bacterial vaccines for poultry in antimicrobial resistance’ OR ‘summary of commercially available vaccines for poultry’ will generate more suitable article for the journal. In addition, there is confusion between prophylactics and vaccines in this review. The contents of section 6 contradict to the description in section 10.

Author Response

We would like to take this opportunity to thank the reviewers for their time and effort put into the review. The comments have helped to improve the revision significantly, and we humbly hope that the revision will be able to respond to the comments made by the reviewers adequately.

Comments and Suggestions for Authors

Reviewer 2:

This review describes the current knowledge of bacterial vaccines combating antimicrobial resistance in poultry. This is very well written review and will provide useful summarization of current status of the field. However, this review touched too many aspects of bacterial vaccines for poultry to provide any beneficial information for the expert readers. This review would provide very little useful information to the reader of the journal “Vaccines”.

Response: Thank you very much for your valuable comments and suggestions to improve the manuscript.

  • More detailed review of ‘the role of bacterial vaccines for poultry in antimicrobial resistance’ OR ‘summary of commercially available vaccines for poultry’ will generate more suitable article for the journal.

Response: Thank you very much for your valuable comment. We have added a section (section 9) and a table (Table 2) that narrated the summary of commercially available vaccines against various bacterial diseases in poultry. And the role of bacterial vaccines for poultry in reducing antimicrobial resistance has already been discussed in section 11. We also provided a figure for clarification.

  • In addition, there is confusion between prophylactics and vaccines in this review.

Response: Prophylactics is a broad term covering many things, including medicine and vaccine. The vaccine could be considered prophylactic, e.g., immunoprophylactic, so there should not be any confusion.

  • The contents of section 6 contradict to the description in section 10.

Response:  Thank you very much for pointing out this. We have now revised section 6 (now section 7) for clarification.

Best regards,

Md. Tanvir Rahman (Corresponding Author) DVM, MSc (Canada), Ph.D. (UK), Postdoc (Germany)
Director, 
Professor Muhammed Hussain Central Laboratory, Professor of Microbiology,  Department of Microbiology and Hygiene,
Faculty of Veterinary Science, 
Bangladesh Agricultural University, Mymensingh-2202, Bangladesh. Phone. + 88-01913323307; Fax + 88-09161510 E.mail: [email protected]

Round 2

Reviewer 2 Report

Revision responded the critique satisfactorily